# Effect of Depth Band Replacement on Red, Green and Blue Image for Deep Learning Weed Detection

**DOI:** 10.3390/s25010161

**Published:** 2024-12-30

**Authors:** Jan Vandrol, Janis Perren, Adrian Koller

**Affiliations:** Institute of Mechanical Engineering and Energy Technology, Lucerne University of Applied Sciences and Arts, CH-6048 Horw, Switzerland

**Keywords:** RGBD, YOLO8, object detection, weed detection, deep learning, band substitution

## Abstract

Automated agricultural robots are becoming more common with the decreased cost of sensor devices and increased computational capabilities of single-board computers. Weeding is one of the mundane and repetitive tasks that robots could be used to perform. The detection of weeds in crops is now common, and commercial solutions are entering the market rapidly. However, less work is carried out on combatting weeds in pastures. Weeds decrease the grazing yield of pastures and spread over time. Mowing the remaining weeds after grazing is not guaranteed to remove entrenched weeds. Periodic but selective cutting of weeds can be a solution to this problem. However, many weeds share similar textures and structures with grazing plants, making their detection difficult using the classic RGB (Red, Green, Blue) approach. Pixel depth estimation is considered a viable source of data for weed detection. However, systems utilizing RGBD (RGB plus Depth) are computationally expensive, making them nonviable for small, lightweight robots. Substituting one of the RGB bands with depth data could be a solution to this problem. In this study, we examined the effect of band substitution on the performance of lightweight YOLOv8 models using precision, recall and mAP50 metrics. Overall, the RDB band combination proved to be the best option for YOLOv8 small and medium detection models, with 0.621 and 0.634 mAP50 (for a mean average precision at 50% intersection over union) scores, respectively. In both instances, the classic RGB approach yielded lower accuracies of 0.574 and 0.613.

## 1. Introduction

Many grazing pastures suffer from weed issues. The low value creation on these fields prohibits expensive weed management activities, which leads to pastures that are often untended, with progressively larger outgrowths of plants which farm animals do not eat or are of a low nutritional value. Furthermore, invasive species may become more easily entrenched. Even when the part of the plant that is above the ground is cut, the roots can be strong enough to continue growing. Mowing the whole pasture after grazing may also be suboptimal, because over time, this favors vegetatively over generatively procreating plants. While this may increase the nutritional value (protein and total digestible nutrients), it will reduce the fiber and structure of the forage, which are necessary for a healthy animal gut [1].

The periodic defoliation of broad-leaf dock—a particularly persistent weed—was shown to weaken the plants enough to keep them in check [2]. Robot-based defoliation of these weeds would require a lightweight, computationally affordable detection solution that can be fitted onto compact rovers and, with high accuracy, distinguish between grazing plants and weeds.

The automated discrimination between weeds and grazing plants is challenging, as they often share similar textures and structures. Several studies [3,4,5] outlined the following difficulties with weed detection: occlusion, similarity in color and texture, plants being shadowed in natural light, color and texture variations due to lighting conditions, similarities in the structures of different species and changes in plant structure during different growth phases. Originally, weed detection was solved in multiple stages—preprocessing, vegetation extraction, feature extraction and classification [4]. However, advancements in both hardware and software solutions have given rise to the employment of deep learning (DL) techniques in weed detection [5,6].

DL is a subdiscipline of Machine Learning (ML) that uses multiple layers of neural networks to extract increasingly more abstract information from input data. In the context of image processing, this can mean moving from object edges to object structures. Thanks to this, DL approaches have an advantage over previously used methods. The similarity between plants can be overcome by the robust learning capabilities of DL. Agriculturally focused studies do indeed report better performance of DL algorithms over traditional approaches [4,7].

With the superior capabilities of DL methods in terms of recognizing structure and texture, some of the issues in weed detection can be addressed. There are, however, still issues of variable lighting conditions. To alleviate this, another source of information apart from the usual RGB data could be used: depth. Depth data—the distance of objects from a sensor—can be obtained using various means. Sensors using structured light, time of flight or vision methods exist. In the context of this paper, vision is the most logical choice, since RGB data need to be collected. While it is possible to obtain depth estimates using monocular cameras [8], a stereo camera is a more robust option. Stereo cameras have become smaller and cheaper in recent years, to the point where there are only a few advantages to using monocular cameras in autonomous tasks. Depth has several beneficial attributes for weed detection. It is fully invariant to illumination and color changes [9]. It also provides additional information about plant structures. Additionally, the weeds in the pastures will be more prominent, as they are not grazed on.

Considering the potential advantage of including depth for weed detection, it would be optimal to include it in the detection process. Unfortunately, the computational requirements for utilizing RGBD information in object detection are higher compared to regular RGB methods. For example, Takahashi et al. [10] reported a 40% decrease in frames per second (from 74 to 44 on a dedicated workstation using GTX 1080Ti) for their RGBD processing method. This leads to significantly higher power requirements, which is detrimental for battery-powered autonomous vehicles.

While RGB bands and depth convey different kinds of information, replacing one of the colors with depth data might yield additional information while having a low impact on the texture patterns. The question then is whether substitution of one of the RGB bands with depth information will lead to a better prediction accuracy while maintaining a lower computational cost. Our paper aims to answer this question with a comparative analysis of object detection results, utilizing different band combinations.

The paper is organized using the following structure: Section 2 contains a review of RGB and RGBD object detection methods and relevant weed detection efforts. Section 3 details the method used to evaluate RGB band substitution in weed detection, with the results being outlined in Section 4. The paper concludes with the discussion in Section 5.

## 2. Related Work

### 2.1. Deep Learning for Weed Detection

Over the years, many methods have been developed for recognizing weeds, be it color transformation [11], segmentation [12], plant morphology [13] or spatial distribution (plant row identification) [14]. DL in weed detection became popular for its ease of use and robustness [7].

Hasan et al. [15] tested several DL architectures on four weed types in corn fields. The DL methods were YOLOv7, YOLOv8 and Faster-RCNN (Region-based Convolutional Neural Networks). The dataset contained 5997 images with augmentations. The results showed that both YOLOv7 and YOLOv8 are capable of high mean average precision values: 89.93% and 89.39%, respectively. Such good performance allows for the use of automated field robots and drones in precision agriculture.

One such example is an object detection system for precision sprayers for strawberry and pea fields, developed by Khan et al. [16]. They collected 2800 images of weeds which were downsampled to a resolution of 448 × 448 pixels and input into a customized detection framework. The framework was based on a Faster-RCNN architecture with a ResNet-101 network and tailored anchor points. The resulting model achieved 95.3% weed detection accuracy.

Another example is the work by Asad and Bais [17], who proposed a weed mapping method with a semantic segmentation step during the labeling process. The segmentation of plants from the background was achieved using maximum likelihood classification. Pixel-level data instead of bounding boxes were then presented to a SegNet DL architecture with a ResNet-50 extractor. The resulting model was able to localize weeds with 82% accuracy.

In the context of weed detection, depth has also been used to improve results. Xu et al. [18] used RGBD data for detecting weeds in wheat fields. The depth image was converted to a PHA (phase, height and angle) image to convey additional information and have the same structure as RGB images. The framework used in this study comprised three detection networks—two specific ones to each image and one shared. The networks were based on Faster R-CNN and a VGG16 architecture. The three outputs were fused using individual weights at the decision level of the algorithm. The study reported a 42.9% mean average precision when detecting broad-leaf weeds.

### 2.2. RGBD in Object Detection

Depth was found to be beneficial in object detection tasks, even in cases where the depth has only been estimated from RGB images [19]. Hou et al. [9] argued that as depth carries different types of information from RGB, it should not be used at the same input level but have its own network for processing. They proposed calculating the horizontal disparity and pixel angle from the depth and combining them into a three-band image to be used alongside RGB.

Unprocessed depth information has also been used in object detection. Chu et al. [20] developed an outdoor surveillance system comprising two sub-networks, one for RGB and one for depth. The outputs of these networks were fused into a cross-modal object feature set. This set was then used in their depth-adaptive anchor generation module (DAGM). Anchors are bounding boxes of objects and are generated during training based on the object scale and aspect ratio. However, identical objects appear smaller when viewed from a distance. This can cause a decrease in accuracy when working with variable depths in images. The use of the DAGM and depth improved the accuracy of object detectors by approximately 2%.

## 3. Materials and Methods

### 3.1. Weed Selection

Broad-leaved dock (*Rumex obtusifolius*) was chosen as the target of this study. It is a highly invasive perennial weed and common in all temperate climate zones [21]. It is capable of withstanding extreme conditions (both drought and cold). Its robust and aggressive root system makes it difficult to eradicate by single mowing, and grazing animals avoid it. It can therefore quickly spread through grassland and decrease its feeding value. These attributes make it a prime target for automated repeated cutting efforts.

### 3.2. Image Acquisition

As of the time of writing, there were no public image datasets of broad-leaved dock containing depth information. A dataset of 588 images was gathered using a ZED2i stereo camera developed by Stereolabs. This camera is capable of capturing stereo images in multiple resolutions, ranging from 2208 × 1242 to 662 × 376. It has a baseline of 12 cm and a focal length of 4 mm, providing a depth range of up to 35 m. Its reported depth error is less than 2 percent for up to a 10 m depth [22]. The field of view is 72° × 44°. The camera was mounted at a constant height of 190 cm above the ground and pointed downwards. This gave us an image area of approximately 2.8 × 1.5 m. The camera was set to a 2208 × 1242 resolution during data collection.

Images of the vegetative, budding and flowering growth stages of the broad-leaved dock were taken to have a representation of different plant structures. Additionally, 10% of the dataset does not contain any weed images to alleviate bias during learning.

### 3.3. Image Preprocessing

First, the depth estimates and RGB images had to be combined. Depth data that are aligned with left or right stereo images can be accessed using Stereolabs SDK. However, the depth map did not cover a whole image area, as it was calculated from both left and right images. Therefore, each image width was reduced by 300 pixels to remove an area where no depth data could be estimated. Then, normalized depth data were added to the image as a fourth band. The images were also resized to a square for use in object detection training. A resolution of 1280 × 1280 was used as a base dataset. It can be accessed online [23]. A resolution of 640 × 640 was used for the actual deep learning training to facilitate inference on lightweight single-board computers. The aspect ratio was maintained by adding black borders on the top and bottom parts of the image.

The images were labeled using Label Studio software (ver. 1.9.0) [24]. A bounding box approach to labeling using YOLO formatting (center-point X and Y coordinates, width and height) was used. Image augmentation was also applied to decrease the chances of detection model overfitting and improve the overall performance, as described in other studies [25,26]. Noise generation and image transformation augmentation were introduced into the dataset, increasing the number of images to 1176. The images were randomly resized and rotated. Afterwards, Gaussian noise was also added. An albumentation library [27] was used for the task. The dataset was divided into training, validation and test sets with a ratio of 7:2:1. A data loader script was created to use different combinations of RGBD bands, virtually creating four different image datasets for object detection training. Figure 1 provides an example of the various combinations.

### 3.4. Object Detection Training

For testing of the effects of depth band substitution, the YOLO (You Only Look Once) detection model was selected. YOLO models have been found to be highly effective in weed detection in other studies [28,29,30,31,32]. YOLO was first developed in 2015 [33] and became the state of the art in many fields. It introduced a shift in the way that objects were detected from a two-shot to a single-shot system. Two-shot detectors (Faster RCNN or Region-based Fully Convolutional Networks, etc.) first predict the locations of objects in the image and then attempt to classify them. Single-shot detectors (SSD, YOLO) do everything in one pass through the neural network. While this approach loses in terms of accuracy, the increase in speed is substantial and allows for the use of real-time detection on a wide array of devices.

YOLO uses a single CNN for detection. It divides the image into a grid. For each grid cell, the bounding boxes and class confidence are calculated simultaneously. Then, a non-maximum suppression algorithm removes all but the strongest candidates, thus eliminating redundant or erroneous detections.

YOLO was released as an open-source project. Its architecture and algorithms have been iterated multiple times, creating a range of versions. At the time of writing, YOLOv9 has been released. However, in this project, YOLOv8 [34] was used. It comes with various improvements in architecture, anchor-free detections, new convolutions and mosaic augmentations. Overall, it shows an increase in performance (both speed and accuracy) when compared to its predecessors. YOLOv8 provides different model versions with increased complexity:YOLOv8n—3.2 million parameters;YOLOv8s—11.2 million parameters;YOLOv8m—25.9 million parameters;YOLOv8l—43.7 million parameters;YOLOv8x—68.2 million parameters.

Model parameters are directly tied to the computational complexity of its training and inference, as well as its potential to learn patterns, resulting in an accuracy increase. As the larger models are too complex to be run on single-board computers, only v8n, v8s and v8m models were considered. Models were run from scratch, and no pre-trained weights were used. There is a possibility that pre-trained models which did not include plants would decrease the accuracy [35]. The substitution of depth for one of the color bands could also cause issues for the pre-trained model.

To provide comparable results, the hyperparameters were kept constant for all training runs. If changes to hyperparameters were made between runs, it would become challenging to determine the effect of the band substitution on the model’s performance. Table 1 shows the values which were chosen, based on the AdamW hyperparameter optimizer. After training, the results of the model runs were also converted into an ONNX format (Open Neural Network Exchange) to improve the inference speed for RPi5 CPU-only processing.

### 3.5. Evaluation Metrics

The following metrics were chosen to assess the performance of resulting models, based on object detection studies [15,25,28,36]:Precision;Recall;mAP50;Inference speed.

Precision and recall are common metrics for the evaluation of prediction and classification models [37]. Precision is the percentage of correct detections. It measures the ability of a model to select only desired objects. On the other hand, recall is the percentage of found objects. It shows how well a model can find all desired objects in a dataset. These metrics are usually used together. Apart, they hold limited value. Individually, each metric can achieve a high score. High precision can be achieved by only accepting weed detections of extremely high certainty and removing the rest. High recall can be attained by detecting every plant in the image. Therefore, precision and recall are evaluated together and usually have an inverse relationship (as the number of found objects increases, so does the number of false positives, thus decreasing the precision). This trade-off between precision and recall is depicted using a precision–recall curve (PRC). The curve is a plot of precision as a function of recall. As the object detection thresholds are relaxed and more objects are found in the image (increasing recall until all are found), the progression of precision is mapped to it. A good detector will maintain high precision levels for longer.

The mAP50 stands for mean average precision at a 50% intersection over union (IoU). It is often used as a main metric in object detection studies (e.g., the Pascal Visual Object Classes Challenge [36]). The mean average precision is calculated using the PRC. The recall value range (from 0 to 1) is divided into 11 points, and precision is calculated for each of them. The mean of these values creates the mAP metric. The IoU is a bounding box overlap metric. The ground truth object bounding boxes and predicted bounding boxes are compared, and the overlap percentage is calculated. For the calculation of mAP50, only detected objects with at least 50% overlap are considered positive.

The inference speed is the amount of time a model needs to process one image. It is tied to the complexity of the models. The main reason for slower inference is the number of parameters (number of weights and biases) that are included in the model.

### 3.6. Hardware

The object detection training was performed using Google Colab service with 12 GB RAM, NVIDIA Tesla T4 GPU (Nvidia Corporation, Santa Clara, CA, USA) and CUDA version 12.2. Inference speed testing was carried out using Raspberry Pi 5 (RPi5; Raspberry Pi Foundation, Cambridge, UK) with 8 GB RAM and Jetson Nano (Nvidia Corporation). The RPi5 was chosen as a lightweight CPU-based tool for processing image data which can be used on small mowers. The RPi platform has previously been used in weed detection studies [38,39]. While lacking a dedicated graphics card, its power consumption is considerably lower than some of the computer vision kits (e.g., Zed Box—5A19V vs. 5A5V). Jetson Nano is another example of a single-board computer. Unlike RPi5, it has a dedicated GPU, which improves its DL processing times. Jetson Nano consumes 10 W by default but can be switched to a 5 W power-saving mode.

The following software packages were installed:Python 3.11;YOLOv8.1.2;PyTorch 2.2.2;CUDA 12.1 (Jetson only);Additional dependencies.

## 4. Results

### 4.1. Model Accuracy

Table 2 provides an overview of the precision, recall and mAP50 metrics across different models. For the v8n model, the mAP50 scores of RGB (0.557) and RGD (0.56) were closely tied. However, RGD had higher precision and recall scores, suggesting that the boundary box estimates were poorer. When examining the PRC for YOLOv8n (Figure 2), RGB and RGD both started similarly. In the middle ranges, the RGB suffered a drop in precision, but it performed better at higher recall values. The RGD had the opposite tendency. Therefore, RGD might perform better in cases where the detection thresholds are higher. The DGB variant had the highest precision score of 0.632 but suffered from the worst recall of all with 0.512, implying that while it can correctly reject erroneous object proposals, it struggled with finding all weed instances in images. Both DGB and RDB had similarly low mAP50 scores: 0.533 and 0.532.

The YOLOv8s results (Figure 3) differed from those of YOLOv8n. In this case, the RGB was the worst-performing band combination in all metrics. The PRC showed that it was outperformed by all other versions, with a recall of 0.3 to 0.7. The best performance for this model was that of the RDB band version. Its mAP50 score was increased by 8% compared to RGB (0.621 vs. 0.574), precision by 13% (0.635 vs. 0.562) and recall by 9% (0.572 vs. 0.523). Its PRC also showed a solid performance in all ranges. In this model, RGD came second, with an mAP50 of 0.596. Its recall of 0.619 was higher than that of RDB but suffered from the second lowest precision score of 0.586.

In the case of YOLOv8m, RDB had the highest mAP50 score of 0.634 and a precision of 0.692. Its PRC (Figure 4) showed a solid performance until the recall value of 0.7, where it fell in line with RGB. The RGB was the second best performing this time, with an mAP of 0.613, which is only 3% behind RDB. The precision score (0.64) was 7.5% lower than that of RDB. Interestingly, RGD showed a much higher recall value compared to the other band variants. At 0.632, it was 11% higher than the next value of 0.566 for RGB and 23% higher than the lowest value of 0.512 for DGB.

### 4.2. Inference Speed

The evaluation of inference times on RPi5 and Jetson Nano is provided in Table 3. The measured speed is in milliseconds. The number is the average of the inference times of all images in the testing dataset.

The RPi5 performs well with the v8n model, only needing 431 ms to perform inference. This speed would allow for a continual survey of the fields without pause. While v8n has the worst performance in terms of accuracy, it might be considered in cases where the field robot is constructed with higher movement speed in mind. Since the inference time of the next model (v8s) is 808 ms—an increase of 87%—the trade-off in accuracy can be acceptable. However, depending on the field robot design, camera placement, field of view and operating speed, v8s can also be considered viable. As long as there is enough overlap between consecutive images to avoid detection errors from a lack of data, the increase in accuracy of v8s over v8n will provide better weeding results. The v8m model, on the other hand, is not practical for field operations with a 1835 ms processing time. If higher accuracy is critical for field operations, different hardware should be considered.

As expected, the dedicated GPU of Jetson Nano allows for faster object detection processing times than RPi5. The v8n model inference is only 176 ms. That is a 59% decrease in the required time. Considering the movement speed of small field robots, the performance can be counted as real-time processing. The v8s processing time is also fairly low. At 342 ms, it is 26% faster than the performance of RPi5 with the v8n model and can be considered fully viable. The v8m model is more cumbersome, with 758 ms. However, in terms of performance, it is still 50 ms faster than the RPi5 using v8s.

The choice between RPi5 and Jetson Nano also comes down to project requirements, practicality, user experience and choice of peripherals. For robots with higher operating movement speeds, Jetson Nano should be considered. However, RPi has much wider support for sensors and other peripherals, as well as a larger user base and support.

## 5. Discussion

This study showed there is potential in depth band substitution. Based on the results, it appears the RDB combination could be beneficial for weed detection in pastures. It proved to be a superior option in two out of the three tested models. While its mAP50 score in the v8n model was lower, it had better precision and recall scores than RGB. This points to an issue with correctly localizing plants, which could potentially be attributed to insufficient training data. The dissimilarities in plant stages of broad-leaved dock could also play a role. Some studies expand the roster of weed classes with different plant stages [40]. Such a distinction might improve future analyses.

The performance of detection across the tested models is only significant between v8n and v8s. The difference between the best-performing band variants (RGD and RDB) is 6.1%. However, when comparing the performance of RDB in v8s and v8m, the change is only 1.3%. On RPi5, the switch from v8s to v8m would require over a second of additional time to process, while Jetson Nano would need an extra 416 ms. Considering the increase in computational cost, it seems that v8m is not a practical option.

Future studies should focus on expanding this comparative analysis to other object detectors. Different architectures could have divergent responses to the depth band substitution. A more receptive model may exist. Furthermore, additional image data would facilitate a more comprehensive analysis. Another avenue of research is determining how image augmentation changes for images with a depth band. Color transformation especially might have a different effect than with classic RGB imagery. Finally, the question is whether a different product of depth vision might prove more beneficial. In RGBD object detection, disparity and angle data are also used. One of these could provide better results than depth. Potentially, a combination of two depth products with a single color band could also be explored.

## Figures and Tables

**Figure 1 sensors-25-00161-f001:**
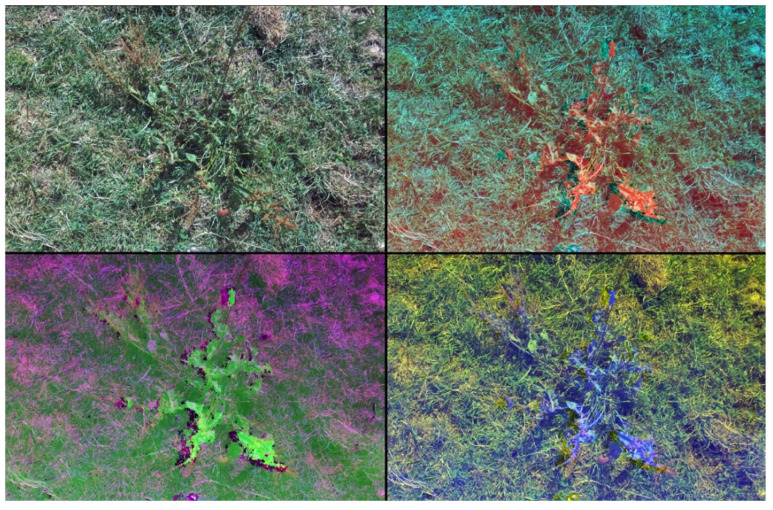
Result of different substitutions of RGB channels with depth. Combinations are as follows: top left—RGB; top right—DGB; bottom left—RDB; bottom right—RGD.

**Figure 2 sensors-25-00161-f002:**
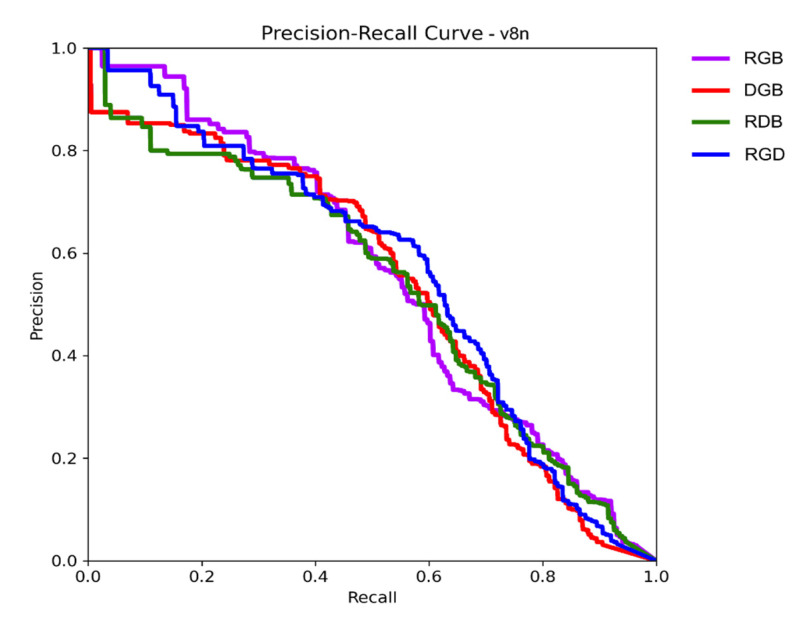
Precision–Recall Curve for v8n model.

**Figure 3 sensors-25-00161-f003:**
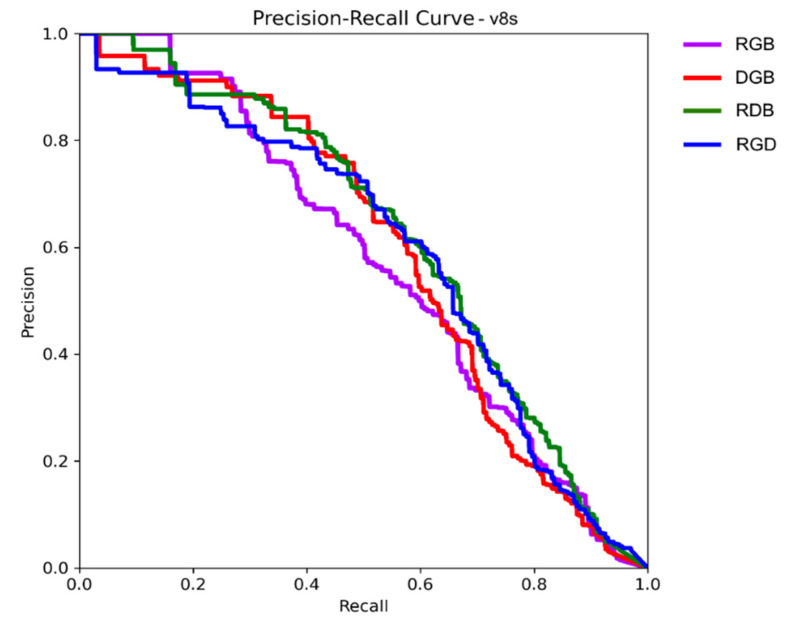
Precision–Recall Curve for v8s model.

**Figure 4 sensors-25-00161-f004:**
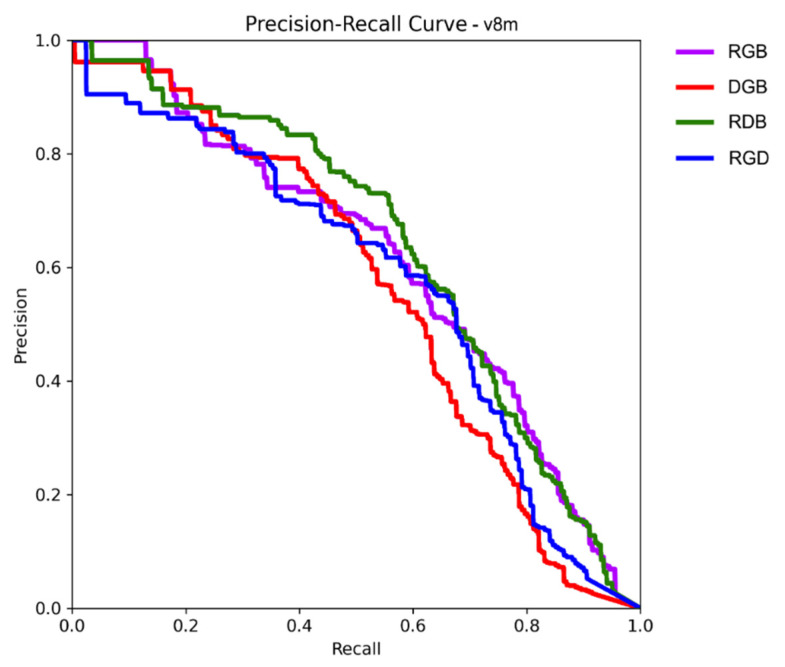
Precision-Recall Curve for v8m model.

**Table 1 sensors-25-00161-t001:** Hyperparameter values used in model training.

Hyperparameter	Value
Initial learning rate	0.002
Final learning rate	0.1
Momentum	0.9
Weight decay	0.0005
Training epochs	300
Batch size	32

**Table 2 sensors-25-00161-t002:** Precision, recall and mAP50 metrics for all band combinations and selected models.

Model	Bands	Precision	Recall	mAP50
v8n	RGB	0.565	0.53	0.557
	DGB	0.632	0.512	0.533
	RDB	0.566	0.557	0.532
	RGD	0.624	0.562	0.56
v8s	RGB	0.562	0.523	0.574
	DGB	0.615	0.564	0.592
	RDB	0.635	0.572	0.621
	RGD	0.586	0.619	0.596
v8m	RGB	0.64	0.566	0.613
	DGB	0.639	0.512	0.562
	RDB	0.692	0.562	0.634
	RGD	0.561	0.632	0.573

**Table 3 sensors-25-00161-t003:** Average inference speed in milliseconds for RPi5 and Jetson Nano single-board computers.

Model	Speed (ms)	
	RPi5	Jetson Nano
v8n	431	176
v8s	808	342
v8m	1835	758

## Data Availability

The data associated with the work described in this publication are available at the Dryad data repository [23].

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
