# Peer review of "Effect of Depth Band Replacement on Red, Green and Blue Image for Deep Learning Weed Detection"

_sensors, 2024, doi:10.3390/s25010161_

Round 1
Reviewer 1 Report
Comments and Suggestions for Authors
Dear authors,
Congratulations on the effort you put into writing this article. Your effort is noted, but some aspects need improvement:
- Be careful when using abbreviations. They should be explained at first appearance, even if they are in abstract and familiar terms.
- Pay attention to the Bibliography; you have some bibliographical items older than 5 years. It would be best if you revised the Bibliography
- After revising the Bibliography, you will need to update the text. For example, the paragraph on page 3, lines 122-128, may not need to be added.
- Beware that something has happened with the formatting. In the Bibliography, the first page is 12 of 15, and the next is 2 of 15.
- On page 4, line 186, you have: "LINK TO DATASET HERE". What's there?
- On page 5, lines 201 to 205 can be found in the text.
- On page 5 - you have inserted Figure 1, but nowhere in the text is it explained or referred to what is there.
- Pages 8, 9, and 10—First, explain the figures and then introduce them so the reader can follow the logical thread. This will help the reader understand the context and relevance of the figures. Also, figures 2, 3, and 4 can be shrunk down a bit so that the article looks neat.
Author Response
Thank you very much for your time to review our work. Please find below our replies:
- Comment1: Be careful when using abbreviations. They should be explained at first appearance, even if they are in abstract and familiar terms.
-->REPLY: We reviewed all abbreviations and introduced full names at first occurrance in the text.
- Comment 2: Pay attention to the Bibliography; you have some bibliographical items older than 5 years. It would be best if you revised the Bibliography
--> REPLY: We reworked the bibliography. However, some of the older references are what we consider "benchmark" papers that justify the approach taken here. Out of the 12 reference older than 5 years, we have removed 5 which were not crucial to our paper's structure.
- Comment 3: After revising the Bibliography, you will need to update the text. For example, the paragraph on page 3, lines 122-128, may not need to be added.
--> REPLY: Yes, thank you for pointing this out. We removed this short paragraph, including the reference along with several other paragraphs pertaining to older bibliography items.
- Comment 4: Beware that something has happened with the formatting. In the Bibliography, the first page is 12 of 15, and the next is 2 of 15.
--> REPLY: corrected (section break in the document removed)
- Comment 5: On page 4, line 186, you have: "LINK TO DATASET HERE". What's there?
--> REPLY: to fulfill requirements from our funding body, we used Dryad as our data repository. They required a manuscript number before we can submit the data (hen and egg problem...). We have now inserted the reference to the dataset. However, the dataset will only be published once the paper is accepted.
- Comment 6: On page 5, lines 201 to 205 can be found in the text.
-->REPLY: Modified the text, reads better now. Thank you for pointing it out.
- Comment 7: On page 5 - you have inserted Figure 1, but nowhere in the text is it explained or referred to what is there.
--REPLY: Referenced Figure 1 in the text.
- Comment 8: Pages 8, 9, and 10—First, explain the figures and then introduce them so the reader can follow the logical thread. This will help the reader understand the context and relevance of the figures. Also, figures 2, 3, and 4 can be shrunk down a bit so that the article looks neat.
-->REPLY: restructured the section "Results" such that each figure is first discussed and then presented in the text. The figures have been resized a bit, however, we find if they are even smaller the performance details become hard to read.
Reviewer 2 Report
Comments and Suggestions for Authors
With great interest I have read the manuscript “Effect of depth band replacement on RGB image for deep learning weed detection”. The authors give an adequate description of the topic of research and the state of the art. The methodology used, by utilizing one of the color bands for storing depth data and using that as input to DL models, is interesting, and the results show that this approach can be promising for automated weed detection as well as many other possible applications.
I found no places where improvement is necessary, except for a minor point mentioned below. Therefore my recommendation is publish with minor revision. I wish to congratulate the authors with this nice publication.
Minor change
Line 186 is supposed to contain a link to a dataset, but this link is not correctly included. This also relates to the statement in lines 401-404, which has not been filled in yet by the authors.
Author Response
Comment 1: Line 186 is supposed to contain a link to a dataset, but this link is not correctly included. This also relates to the statement in lines 401-404, which has not been filled in yet by the authors.
-> REPLY: to fulfill requirements from our funding body, we used Dryad as our data repository. They required a manuscript number before we can submit the data (hen and egg problem...). We have now inserted the reference to the dataset. However, the dataset will only be published once the paper is accepted.